# Exploring Dental Health and Its Economic Determinants in Romanian Regions

**DOI:** 10.3390/healthcare10102030

**Published:** 2022-10-14

**Authors:** Andor Toni Cigu, Elena Cigu

**Affiliations:** 1Faculty of Dental Medicine, Apollonia University of Iași, 700511 Iași, Romania; 2Faculty of Economics and Business Administration, Alexandru Ioan Cuza University of Iași, 700505 Iași, Romania

**Keywords:** dental medical services, Oral Health System, socio-economic development

## Abstract

Sustainable dental health is reflected in the high quality of the medical act and the high quality of the medical service, which cannot be achieved without considering the existing social context, especially the economic development of a state, where certain economic variables can become real levers of influence. The goal of this paper is twofold—theoretical and empirical. Firstly, at the theoretical level, we provide the context and the development of the health legal framework and the state of the Oral Health System and the provision of dental medical services in the eight Romanian Regions of Development. The second aim is to evaluate the relationship between dental health and well-being for the case of regions of Romania over the period 2001–2015. To review the dental health care in Romania, we will use descriptive analysis as the methodology, and to explore the relationship between dental health and economic determinants, we will use an econometric model, the OLS model. Our working hypothesis is that dental health care is influenced by the economic variables in a country. The results show a positive and significant relationship between dental health care and the most important indicator of well-being, the level of income. Of course, an important role is played by the complexity of education, expressed by research and development, which determines a significant positive relationship with dental health in the development regions of Romania.

## 1. Introduction

Medical services show permanent changes because of multi-factor interventions, among which we highlight technological evolution, health policies, demographic changes, etc. In the case of Romania, the health policies and, hence, the quality and quantity of medical services recorded major changes during the period 1990–2022. One of the medical services with a sharp evolution is dental medicine, because there is a greater recognition that oral health has important spillover effects on physical health more generally. Additionally, there are both economic and medical developments of good oral health. Dental health can be appreciated fundamental in managing the whole health system because it allows for the proper nutrition of the body through proper chewing, and it is an important psychological contributor—having healthy teeth gives a pleasant appearance and develops proper self-confidence.

From the perspective of literature, oral health is a determining factor in the sustainability of human capital. Michaela Grossman [1] declared that health is a durable capital good which is inherited and depreciates over time. From the perspective of economic appreciation, the author [1] considers that investment in health takes the form of medical care purchases and other inputs, and depreciation is interpreted as the natural deterioration of health over time. Therefore, from an economic and social perspective, state authorities are challenged to develop sustainable health policies based on public health spending, a health insurance system with permanent access to health services and medical education, all of which contribute to sustaining individual health as human capital and the main resource of sustainable development. Of course, the field of health includes oral health and, especially, dental health care. Grossman [1] developed an economic model of individual health behavior or demand for health and found that individual behavior is determined by the balancing of the benefits and opportunity costs of health change. The Grossman model of health demand [1] provides a way of understanding individual behavior and emphasizes the need to design healthy public policies in ways that respond to the varying contexts and circumstances of individuals. Glied and Neidell [2] found that the impact of the oral health status on earnings differs between women and men.

Dental health policy is one of the branches of health policy, and, therefore, the interaction with the economic dimension is inevitable. Lalloo, Myburgh and Hobdell [3] found that there is a relationship between the level of socio-economic development and dental caries, where dental caries can be considered a good proxy measure for socio-economic development.

Few studies have tested for the existence of supplier-induced demand in dental markets or a positive correlation between the number of dentists in an area and the utilization of dental care [4,5,6]), on the assumption that more dentists generate more demand [7]. Trohel et al. [8] consider that oral health inequalities based on socio-economic characteristics can be identified in many countries. The determinants include educational level, occupational background, income [9,10,11,12,13,14] and place of residence [15,16]. In the opinion of Costa et al. [17], the most frequent socioeconomic indicators associated with caries are schooling, income and socioeconomic status. Steel et al. [18], based on multiple linear or logistic regressions, concluded that income sometimes has an independent relationship with inequalities, but education and area of residence are also contributory. The authors [18] consider that appropriate choices of measures in relation to age are fundamental for understanding and addressing inequalities.

Our paper aims to present empirical evidence on the relationship between dental health and the well-being of the individual for the eight Romanian Regions of Development without legal personality, as established by law over the period 2001–2015, using a linear regression model. In the first part of the paper, we review the literature and we present the status of the dental health care system in Romania based on a legal framework and the provision of dental medical services. The second part of the paper will be focused on empirical evidence based on the econometric model. 

Our hypothesis is that dental health care is influenced by the economic status of the country. Thus, the more developed the state, the better founded and implemented health policies are, and the more willing citizens are to pay attention to their overall health, including dental health. In this context, the more developed the state, the more citizens will be willing to give up part of their income for the maintenance of dental health by paying for their dental health care. 

The novelty of the paper is given by the fact that we study the relationship between dental health and a series of economic indicators in the eight regions of Romania, and, to our knowledge, so far, the methodology used and this relationship have not been studied.

The paper is structured as follows: Section 2 provides a review of dental health in Romania; Section 3 describes the method, variables and data sources; Section 4 summarizes the results of the empirical study conducted in Romanian regions; Section 5 discusses the results and how they can be interpreted from the perspective of previous studies and the working hypotheses. The paper ends with conclusions and references.

## 2. Review on Dental Health Care in Romania

The legal perspective of the dental medical system and oral health in Romania is a complex one, which presupposed an evolution in time and which represents a process in continuous development. What is certain is that the very fundamental law of Romania [19] establishes, in Article 34, the right to health care, which is guaranteed for every citizen; furthermore, the state is obliged to take measures to ensure hygiene and public health.

In 1990, the Romanian government embarked on a fundamental, albeit slow-paced, health care reform, shifting the health system towards a more decentralized and pluralistic social health insurance system, with 18 health systems and Romanian contractual relationships between health insurance houses as purchasers and health care providers. 

The main legislative acts were introduced between 1995 and 2002. In 2006, these were replaced by the Law 95/2006 on Health Care Reform [20], which is still in place today. This harmonized the national legislation with the acquis communitaire. In this context, Law 95/2006 [20] is considered the key legal act governing Romanian health, which consists of 19 titles that cover almost all aspects of the health sector, including (i) its financing, organization and governance; (ii) the provision of services encompassing public health, primary care, emergency care, specialized outpatient care, hospital care and pharmaceutical care; (iii) the consideration of health care professionals’ practice; (iv) the coordination and harmonization of the Romanian social health insurance system with the social health insurance systems of other EU Member States. The Law 95/2006 [20] has been subject to over 1300 amendments since it came into force in 2006. 

In 2012, a new health law was proposed, following numerous studies that had shown poor performance of the health system, along with mounting pressure from the general public and health professionals. The proposal envisaged replacing the system of controlled resource allocation with regulated competition at both the health insurer and service provider levels (after the Dutch model), introducing the ‘money follows the insured’ principle and modifying the structure and functioning of service providers to enable service integration and improved continuity and quality while also ensuring cost efficiency. The proposal was rejected amidst protests and calls for the resignation of the president. The most recent reforms focused mainly on introducing cost-saving measures.

The main institutions at the Romanian national level are the Ministry of Health, the National Health Insurance House and the professional organizations. The Parliament has a key position in the policy process, representing the legislative power and controlling the activities of the government. The Ministry of Public Finances oversees the financial resources raised for and spent on health care and plays a key role in decisions on health sector reforms when they involve changes in public finances. The Court of Accounts controls the formation, administration and utilization of state financial resources in the public sector. The Ministry of Transport, Ministry of National Defense, Ministry of Internal Affairs, Ministry of Justice and Romanian Intelligence Agency also play a role in the health system by operating their own parallel health systems, as well as through intersectoral cooperation.

In order to understand Romania’s status, we will carry out Romania’s Human Development Index (HDI) [21] (Figure 1), which places the human individual at the center of all developmental flows, being a composite index measuring the average achievement in three basic dimensions of human development—a long and healthy life, knowledge and a decent standard of living. An HDI value closer to 1 indicates very high human development, and an HDI value closer to 0 reflects low human development.

According to the HDI, Romania ranks 49th out of 66 countries at the Very High Human Development level, with a minimum of 0.701 in 1990 and maximum of 0.828 in 2019. Every year, Romania’s position in the ranking increases, positioning itself better and better, but this positioning is dependent on the human development paradigm which emphasizes two simultaneous processes—firstly, the building of human abilities and how people use them to function in society and, secondly, making choices between options that they have in all aspects of their lives [22]. 

One of the relevant indicators that justifies the status of health care in Romania is the current health expenditure (% of GDP) in Romania (Figure 2). The Romanian health system is financed from four main sources, namely, national health insurance funds as the most important source of revenue for health (almost 67%), the state budget, local budgets and OOP (out-of-pocket) payments as the second source of revenue for health.

Romania has a lower share of its national wealth devoted to health care than other countries. The share of health in GDP fluctuated between 1995 and 2019, where the minimum level was 1.9% of GDP in 1997 and the maximum was 5.0% of GDP in 2019. The European average is around 9%. Of course, with the onset of the SAR-CoV-2 pandemic, all states have been challenged to allocate additional funds to the health care system for proper management and to avoid bottlenecks, but for now, these data are not available at the moment.

With regard to dentistry, the functioning of the dental offices is performed based on the Norm of 31/07/2007 regarding the structure and operation of the medical and dental offices, published in the Official Gazette, Part I no. 575 of 22/08/2007 [24]. Additionally, the National General Assembly of the College of Dentists adopted the Code of Ethics [25] for Dentists, published in the Official Gazette in August 2005. The Code of Ethics [25] contains the rules of moral and professional conduct regarding the exercise of the dentist’s rights and duties. Among the functions of the deontological code, the following are more important: the promotion of a trusting relationship between doctors and patients, the guarantee of the quality of professional training, the promotion of professional-deontological behavior as well as the guarantee of professional secrecy.

The profession of dental technicians is based on Law no. 96/2007 regarding the exercise of the profession of dental technicians, republished on 24/04/2009 [26].

All dentists who wish to practice in Romania should become members of the Romanian College of Dentists or have a temporary or occasional practice notice, having the rights and obligations established by law [20]. The Romanian College of Dentists operates on the basis of Title XIII of Law no. 95/2006 on health care reform [20], republished with subsequent amendments and completions, as well as the Organization and Functioning Regulation adopted by the Decision of the National General Assembly no. 5/2007 [27], with subsequent amendments and completions. The Romanian College of Dentists is a professional, apolitical, non-profit body under public law with responsibilities delegated by the state authority in the field of the authorization, control and supervision of the dentist profession as a liberal profession and authorized public practice with institutional autonomy in its field of competence, as well as normative and professional jurisdiction [28].

The Romanian College of Dentists involves, as its main attributes [28]: (i) the control and supervision of the exercise of the profession of dentists; (ii) the application of the laws and regulations that organize and regulate the exercise of the profession; and (iii) representing the interests of the dental profession and maintaining the prestige of this profession in social life.

In the Romanian territory, almost 26,723 dentists are registered at the RCD (The Romanian College of Dentists) in 2022 (see Figure 3). A total of 6245 are in Bucharest, which is 23.37% of the total number of dentists in the country (almost a quarter). Iași county has 2114 dentists, and Cluj, Timiș, Constanța and Bihor counties have over 1000 dentists, which is 30% of dentists. The remaining 47% of dentists are in the other 36 counties. Practically, more than half of the dentists (53%) practice the profession in only five counties in the country and the Municipality of Bucharest [29].

The comparative approach of the total number of doctors and the number of dentists in the regions of Romania in the period 2001–2015, as well as the ratio to one-hundred-thousand inhabitants, reflects a relatively similar distribution for the Bucharest-Ilfov region, which has the highest number of doctors in both categories (see Figure 4). Bucharest is the capital of Romania and, at the same time, is an important university center with medical education. Bucharest also has the largest population compared to other urban areas of the country. The West Region is better in terms of both categories of doctors relative to one-hundred-thousand inhabitants, and the North-West Region is better in terms of the global number of doctors. The North-East region is in third place in the global number of doctors in both categories, but relative to the population (per one-hundred-thousand inhabitants), it is in sixth place among doctors and in fourth place among dentists. The South-Muntenia region ranks last in almost all categories. 

From the perspective of the way the dentists carry out their activities, dental care is provided by the Individual Medicine Offices (IMO) in general, as can be seen in Figure 5. Overall, dental care represents a fascinating mix of the public and private spheres [30].

The trend of European countries is that dental care is predominantly performed by private medical practices, and this trend is also found in Romania. Most practices are organized in the form of internal medicine practices or in the form of medical clinics. Dental care in Romania is provided through a network of 15,100 ambulatory facilities, most of them private (12,127; 86%), which are organized as private dental practices (11,931) or medical dentists’ civil societies (a form of professional organization for liberal professions in Romania) (196). 

The size of the development is often estimated by GDP per capita by the economists, so Figure 6 reflects the dynamics of this indicator in the period 2001–2015 in Romanian regions.

According to the data in the graph, we can observe the detached and very fluctuating evolution of the Bucharest-Ilfov Region compared to the other regions of Romania. Significant fluctuation, in the sense of decreasing GDP per capita, is registered during the world economic recession (2008–2011). The other regions also register a slight decrease in GDP per capita during the economic crisis, but it is not as marked as in the case of the Bucharest-Ilfov Region. The lowest level of GDP per capita is registered in the North-East Region, with a slight increase towards the end of the period. Four of the regions have a slightly higher GDP per capita than the North-East Region, but they are not very differentiated. The West region is the one with the highest GDP per capita, but the high gap is still maintained in relation to the Bucharest-Ilfov region.

The real growth rate of regional gross value added (GVA) at basic prices (see Figure 7) reflects that the North-East and South-West Oltenia regions recorded the lowest performance in the period 2001–2015. The Bucharest-Ilfov region stands out significantly, registering an accentuated growth.

## 3. Materials and Methods

The paper provides empirical evidence on the relationship between dental health and the well-being of the individual for eight Romanian Regions over the period 2001–2015. For empirical evidence, we used OLS Model (Ordinary Least Squares Model). The eight regions in Romania are not administrative-territorial units and do not have legal personality, being constituted on the basis of agreements made between the representatives of the county councils. The development regions, according to Law no. 315/2004 on regional development in Romania [31], are the framework for the development, implementation and evaluation of regional development policies, as well as the collection of specific statistical data, in accordance with the European regulations issued by EUROSTAT for the second level of NUTS 2 territorial classification, existing in the European Union. The eight regions are as follows: Nord-Vest (North-West), Centru (Center), Nord-Est (North-East), Sud-Est (South-East), Sud–Muntenia (South-Muntenia), Bucuresti-Ilfov (Bucharest-Ilfov), Sud-Vest Oltenia (South-West Oltenia) and Vest (West). For empirical evidence, we used Pooled OLS regression. The chosen period (2001–2015) is justified by the availability of the official databases of EUROSTAT [23].

The novelty of the research is justified by the relationships under study and the choice of the determinants of dental health, as well. Our econometric model is in accord with the literature [32] and is presented below in Equation (1):(1)$$y_{i,t}=\beta_{1}{}_{i,t}+\beta_{2}X_{2i,t}+\beta_{3}X_{3i,t}+u_{i,t}$$
where: *i* refers to the country ($i=\bar{1,\;8}$); *t* refers to the year ($t=\bar{1,15}$); *y* is the dependent variable; X is the independent variable; $\beta_{1},\beta_{2}\;\mathrm{and}\;\beta_{3}$ are the coefficients of the explanatory variables; $u_{i,t}$ are the observation-specific errors.

For the model, the dependent variable is Dental Health (*DentalH*), reflected by the number of dentists per one-hundred-thousand inhabitants, starting from the presumption of the authors [4,5,6,7,30], according to which a greater number of dentists generates more demand and determines the greater attention of the citizens regarding dental health care.

We use, in our paper, four variables as determinants of Dental Health (see Table 1), namely: (a) income of households (EUR/inhabitant) (*Income*); (b) R&D personnel and researchers (% in labor force) (*R&D*); (c) gross fixed capital formation (*Investment*). From an income perspective, the literature identifies its role in dental health management [2,9,10,11,12,13,14,17]. R&D personnel and research include, in addition to research, the idea of education, which is also considered an important factor for dental care [15,16,17]. 

Table 2 and Table 3 provide descriptive statistics for the variables included in the model for the eight Romanian regions regarding the relationship between dental health and the well-being of the individual over the period 2001–2015.

Above is the correlation matrix for all variables in the model. The numbers are the Pearson correlation coefficients that go from −1 to 1. A value closer to 1 means a strong correlation. A negative value indicates an inverse relationship (roughly, when one goes up, the other goes down). According to our data, there are strong correlations between variables, the values being in the interval [0.62–0.92]. All variables show significance at the 0.05 level.

## 4. Results

The results of the regression analysis represent empirical evidence which justifies the purpose of the research regarding the relationship between dental health and the well-being of the individual.

We used the OLS Model for the panel data developed for the eight Romanian Regions over the period 2001–2015, as can be seen in Table 4. To estimate equation 1 and structure the results, we first solved the problems of spurious regression. We used the variance inflation factor (VIF) to check for multicollinearity, showing values of 6.08 for our regressions. The adjusted R^2^-square shows the amount of variance of Y (outcome variable *DentalH*) explained by X (prediction variables *Income, R&D* and *Investment*). In this case, the prediction variables explain 82% of the variance in Dental Health (*DentalH*). This provides a more honest association between X (prediction variables) and Y (outcome variable). We used robust standard errors to control for heteroskedasticity.

Our results show, according to the OLS model provided in Table 4, that the coefficients of the income of households and R&D personnel and researchers are positive (+) and statistically significant, as predicted by our hypothesis. The investment variable is statistically significant but has a negative impact (−). Two-tail *p*-values test the hypothesis that each coefficient is different from 0. In our model, all prediction variable (*Income, R&D and Investment)* are statistically significant in explaining the outcome variable *DentalH* (*** *p* < 0.001).

To complete the image of our model, we want to identify the status of each region in the analysis, as can be seen in Figure 8.

Figure 8 plots the prediction from a quadratic regression, and it adds a confidence interval for the eight regions in Romania. As can be seen, the North-East (Nord-Est) region has a high number of dentists relative to the level of income per inhabitant, the explanation being that the North-East Region has an important university and medical research center in Iasi. The tendency of dentists is to stay in Iasi after graduation and, in general, in the North-East region. In the case of the South-East (sud-Est), South-West Oltenia (Sud-Vest Oltenia) and South-Muntenia (Sud-Muntenia) regions, we notice a very low level of dentists, these also being the regions that do not have university centers in dental medicine. The best location is the Bucharest-Ilfov Region, which has high per capita incomes and also a significant number of dentists.

For greater accuracy in reflecting the role of economic development, we will also make a graphical representation in terms of dentists per one-hundred-thousand inhabitants and GDP per capita, averaged over the period 2001–2015.

Figure 9 plots the prediction from a quadratic regression, and it adds a confidence interval for the eight regions in Romania based on dentists per one-hundred-thousand inhabitants and GDP per capita. As can be seen, the results are similar to those obtained in the previous figure, reflecting that the North-East (Nord-Est) region has a high number of dentists relative to the level of GDP per capita. In the case of the South-East (Sud-Est), South-West Oltenia (Sud-Vest Oltenia) and South-Muntenia (Sud-Muntenia) regions, we notice a very low level of dentists and a low level of GDP per capita. The best location is the Bucharest-Ilfov Region, which has a high GDP per capita and also a significant number of dentists.

## 5. Discussion

The obtained results are in accordance with the literature [2,9,10,11,12,13,14,15,16,17] emphasizing that the economic context influences dental health, at least from the perspective of two important indicators, namely, the income of households and R&D (research and development), which are positive and statistically significant, as predicted by our hypothesis. Therefore, education and personal income will determine the health awareness of the human individual in general. Of course, an educated and wealthy human individual will place a lot of emphasis on dental hygiene, which leads to medical check-ups at the dental office. In this context, the higher concentration of dentists will be identified at the level of a community that is aware of its role at the level of an educated and developed community.

The role of investments is significantly negative according to our model, and the explanation may be that it consists of resident producers’ acquisitions used in processes of production and justifies the technical-professional dimension of the economic individual, which is strictly professional-oriented and determines a negligent behavior in terms of dental health. Too much focus on investment processes, without a correct correlation with the variables of sustainable development, can lead to a negative relationship with health, as evidenced by our model.

Complete answers on the impact of economic policies are not likely to be certain, but, overall, we think the evidence supports the idea that well-defined and -targeted economic policies may help in promoting dental health development. However, there are still a few more steps that need to be taken to define the best legal framework for Romania. Political considerations associated with providing equal access to services and placing a greater priority on the health improvements of specific population groups may be important goals.

These findings are understandable given that certain economic variables can become real levers of influence for dental health care and considering the current stage of the Romanian economy and the stage of each Romanian region. Thus, noting that the income of households is a relevant economic variable that influences dental health care in a region, it is important, from the perspective of this indicator, to insist on the elimination of discrepancies and inequalities between citizens’ incomes. Of course, we are not campaigning for a social vision specific to the socialist ideology, but these inequalities must be reduced precisely through the European vision of eliminating very strong economic inequalities between regions and, in general, inside a country. A healthy economy will generate sufficient income for each family and, thus, guaranteed access to high-quality dental health services. Currently, to mitigate these inequalities and for access to dental medical services, the Health Insurance System is used, which does not consider the effective contribution of the citizen, but it is important to be registered in the Health Insurance System as a citizen. We consider that this form of social contribution is beneficial, but if we consider that dental medicine is carried out mainly through Individual Medicine Cabinets and that not all of them apply for funds from the Health Insurance System, then we can realize the importance of each family having sustainable incomes.

A sustainable economy is also justified by the level of Research and Development (R&D), which, according to our analysis, is positive and statistically significant with dental health care. In this sense, the states are driven to raise the level of R&D from the perspective of the objectives established by the Europe 2030 Strategy because of the status of the current society as a knowledge society. Romania has assumed this objective through the Europe 2030 Strategy and is taking significant steps in this direction.

The strength of our study is the novelty of the study regarding the relationship between dental health and the well-being of the individual for the eight Romanian Regions over the period 2001–2015 using a linear regression model. From this point of view, our analysis is an original one and is absolutely novel for Romania, where, as far as we know, there has been no previous study.

The limitation of this study is the lack of variables (oral health and economic) at the Romanian regional level that could better justify the influences, as well as the lack of recent data as close as possible to the present.

Our future research directions are to extend the analysis by evaluating the relationship between oral health policies and the level of socio-economic development of different developed and developing countries in Europe and all over the world, including even more variables in the econometric model. 

## 6. Conclusions

This study has successfully answered the research paper’s questions, and these findings are in accord with our hypothesis that dental health care is influenced by the economic variables in a country. More precisely, our paper highlights the status of the relationship between dental health and the well-being of the individual for eight Romanian Regions over the period 2001–2015 using a linear regression model. From the path analysis results, it was found that the coefficients of the income of households and R&D (research and development) variables are positive and statistically significant, as predicted by our hypothesis. These two variables, which directly impact dental health, are explicable because education and personal income will determine the health awareness of the human individual in general. An educated individual will always be aware of his oral health status and invest in it, based on sustainable income, given that a job with a higher education, according to the law, is on a higher salary scale. A developed state that emphasizes innovation will build its GDP from areas of activity based on innovation, which are also the strongest revenue generators. 

From the investment perspective, the relationship is negative if it is not accompanied by innovation. In this context, the development of entrepreneurship in its forms, but which does not involve innovation, can lead to a consumption of human and material resources, which determines a guaranteed neglect of oral health. These situations appear in developing or transitioning states, as is the case with Romania. The regions with smart cities will invest intelligently, but in Romania, the regions are not characterized by such a level of development.

## Figures and Tables

**Figure 1 healthcare-10-02030-f001:**
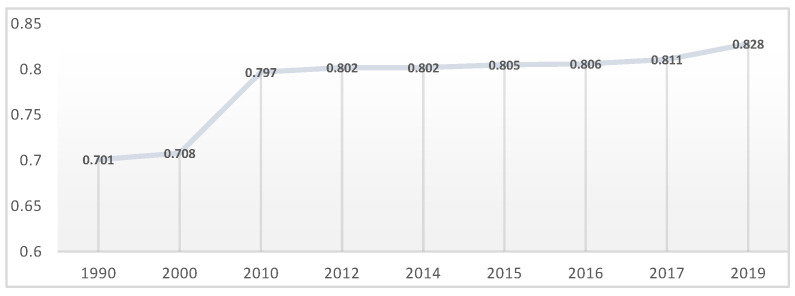
Human Development Index (HDI). Source: computed by authors using HDI [21].

**Figure 2 healthcare-10-02030-f002:**
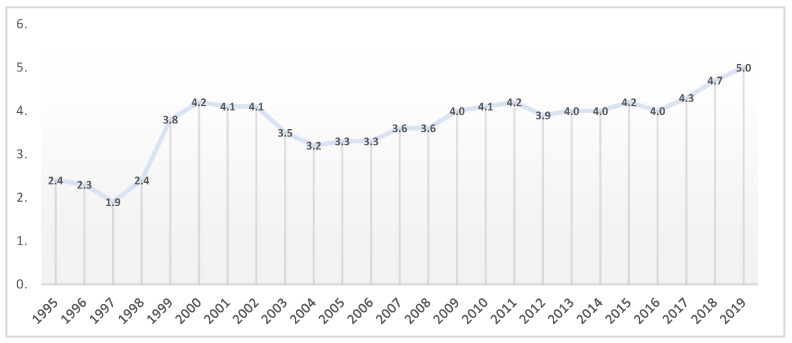
Current health expenditure (% of GDP) in Romania. Source: computed by authors using the Eurostat database [23].

**Figure 3 healthcare-10-02030-f003:**
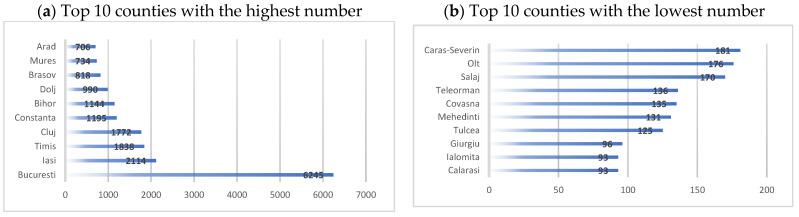
Number of dentists registered by The Romanian College of Dentists. Source: computed by the authors based on the RCD database [28].

**Figure 4 healthcare-10-02030-f004:**
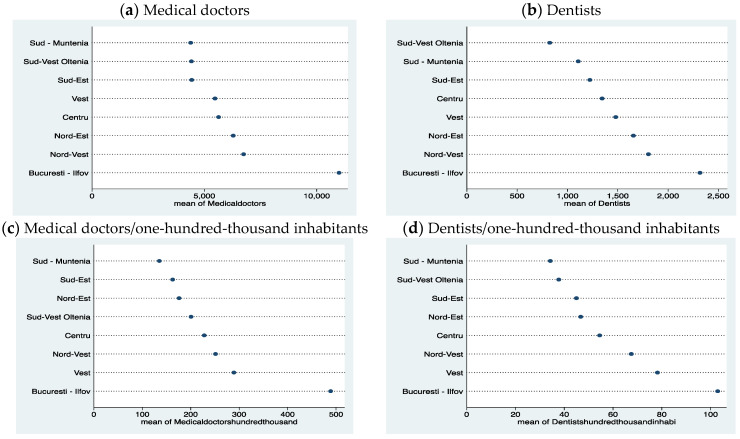
Number of doctors in Romanian Regions. Source: computed by the authors based on the Eurostat database [23] using Stata 15.1.

**Figure 5 healthcare-10-02030-f005:**
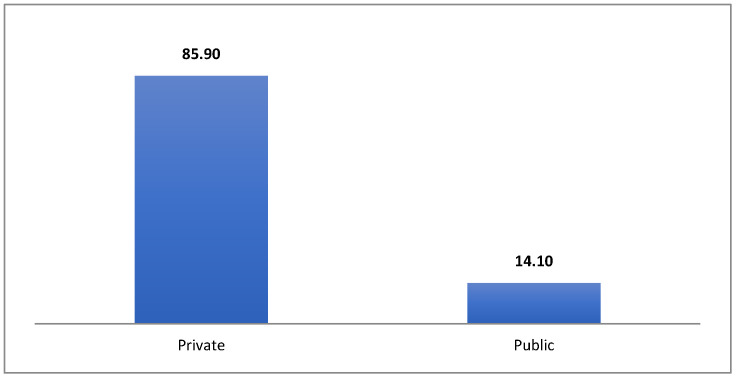
Dental Care Facilities in Romania (%). Source: computed by the authors using the database [28,29].

**Figure 6 healthcare-10-02030-f006:**
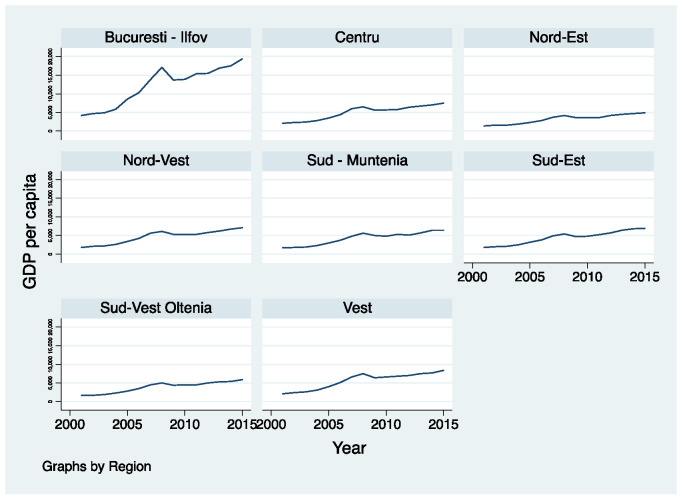
Dynamic of GDP per capita in Romanian regions. Source: computed by the authors based on the Eurostat database [23] using Stata 15.1.

**Figure 7 healthcare-10-02030-f007:**
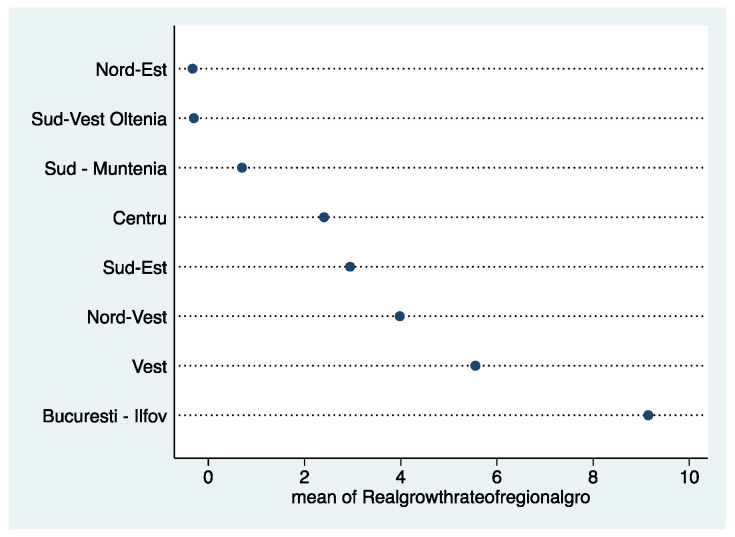
Real growth rate of regional gross value added (GVA) at basic prices. Source: computed by the authors based on the Eurostat database [23] using Stata 15.1.

**Figure 8 healthcare-10-02030-f008:**
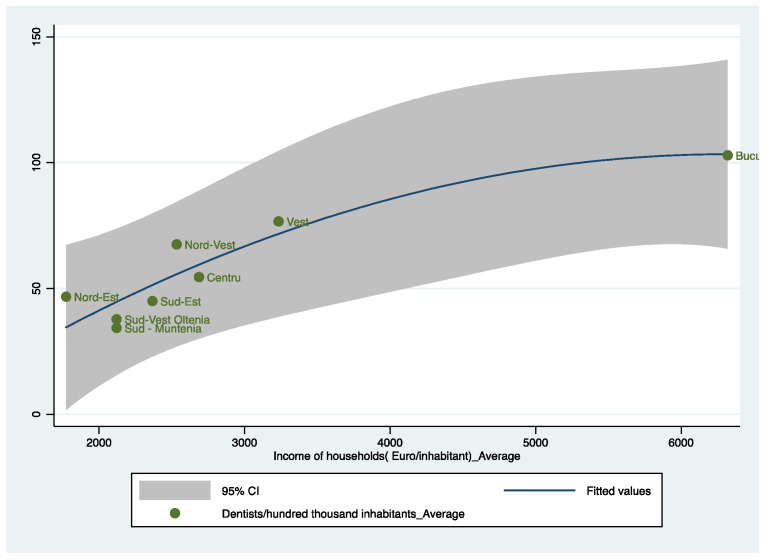
Dentists per one-hundred-thousand inhabitants and income of households (EUR/inhabitant), averaged over the period 2001–2015. Source: computed by the authors based on the Eurostat database [23] using Stata 15.1.

**Figure 9 healthcare-10-02030-f009:**
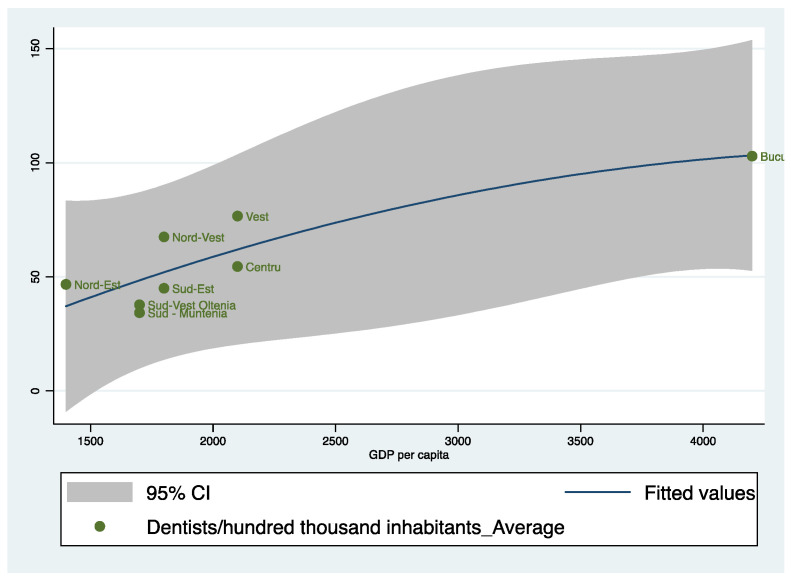
Dentists per one-hundred-thousand inhabitants and GDP per capita, averaged over the period 2001–2015. Source: computed by the authors based on the Eurostat database [23] using Stata 15.1.

**Table 1 healthcare-10-02030-t001:** The variables included in the analysis.

Variable	Definition	Data Source
*DentalH*	Dentists/one-hundred-thousand inhabitants	European Commission [23]
*Income*	Income of households (EUR/inhabitant)	European Commission [23]
*R&D*	R&D personnel and researchers (% in labor force)	European Commission [23]
*Investment*	Gross fixed capital formation	European Commission [23]

Source: computed by the authors, based on sources indicated inside the table.

**Table 2 healthcare-10-02030-t002:** Pairwise correlations.

Variables	(1)	(2)	(3)	(4)
(1) DentalH	1.000			
(2) Income	0.703 *	1.000		
(3) R&D	0.842 *	0.743 *	1.000	
(4) Investment	0.692 *	0.803 *	0.924 *	1.000

* shows significance at the 0.05 level.

**Table 3 healthcare-10-02030-t003:** Descriptive Statistics.

Variables.	Obs	Mean	Std.Dev.	Min	Max	p1	p99	Skew.	Kurt.
(1) DentalH	120	58.36	26	20.6	124.31	21.95	123.14	0.797	2.651
(2) Income	120	0.453	0.542	0.138	2.138	0.143	2.009	2.224	6.183
(3) R&D	120	2894.167	1812.343	900	9800	1000	9100	2.015	7.09
(4) Investment	120	3656.914	3821.709	542.03	22,191.5	597.22	17,171.89	2.746	10.572

Source: the authors’ calculation using Stata 15.1.

**Table 4 healthcare-10-02030-t004:** The results of the regression analysis.

	*(1)* *DentalH*
Income	0.0198 *** (0.00151)
R&D	19.68 *** (2.450)
Investment	−0.00622 *** (0.000685)
_cons	14.81 *** (2.360)
*N*	120
*R* ^2^	0.820

Standard errors in parentheses; * *p* < 0.05, ** *p* < 0.01, *** *p* < 0.001; Source: the authors’ calculation using Stata 15.1.

## Data Availability

Not applicable.

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
