# Peer review of "Exploring Dental Health and Its Economic Determinants in Romanian Regions"

_healthcare, 2022, doi:10.3390/healthcare10102030_

Round 1

Reviewer 1 Report

The manuscript is well written and the topic has been covered in detail. However, there should be modifications in structuring and organizing the texts. Please replace the figures and explanations of figures 8 and 9 in the results section. 

I strongly recommend adding a section on the strengths and limitations of the studies with future directions at the end of the discussion.

Author Response

Dear Reviewer 1,

The authors gratefully acknowledge the constructive comments on our paper. Below we reply to your specific comments in the referee report. For your convenience, we replicate your comments in the report first (in black) followed by our reply (in red). In the revised version of the paper, our changes are found in red.

We highly appreciate your contribution to this study with all your constructive comments!

Best regards,

Andor Toni Cigu

Elena Cigu

Reviewer 2 Report

Firstly, I would like to congratulate the authors on the relevance of the public health theme addressed in the article. After reading, it was clear that they have made an excellent review of the state of dental health care in Romania, a relevant and important issue that effectively reflects the social, economic and cultural development of a country.

There are some issues I suggest to see reviewed:

To enrich the article, I advise a review of the abstract with the following well defined points: state of the problem, objectives, methodology, results and conclusions.

In my opinion, the third point concerning the review on dental health care in Romania is too extensive and detailed which difficults reading. I suggest, without harming of the content, to try to make a synthesis of this subject.

I think it would be very positive and relevant for the article to propose some measures and solutions which, according to the authors, and within the reality of Romania, could improve and deliver oral health care to the most disadvantaged populations.

I would like to see the conclusions more objective and responding clearly to the proposed objectives.

Author Response

Dear Reviewer 2,

The authors gratefully acknowledge the constructive comments on our paper. Below we reply to your specific comments in the referee report. For your convenience, we replicate your comments in the report first (in black) followed by our reply (in red). In the revised version of the paper, our changes are found in red.

we highly appreciate your contribution to this study with all your constructive comments.

 Best regards,

Andor Toni Cigu

Elena Cigu

Reviewer 3 Report

1. Abstract: Kindly objectively describe the results of your research. Further, please present all numerical values, such as coefficients.

2. Introduction: I believe that the Introduction (including “2. Literature review” and “3. Review on dental health care in Romania”) was redundant. It would be easier for the reader to understand if the previous studies and research hypotheses were described in a more focused manner. Further, the differences between the previous studies and what the authors are trying to demonstrate in this study are unclear.

3. Methods: The authors set the “number of dentists per 100,000 population,” the dependent variable, as an indicator of dental health. However, its relationship with other indicators, such as caries, periodontal disease, and the number of patients, is unclear. Is it a good choice to define this variable as the sole indicator of dental health? It seems more appropriate to hypothesize this indicator “to clarify the relationship between the dental care provision system and various economic factors.”

4. Results: P9, L313–315: This sentence is better discussed in the Discussion.

5. Results: In the text, please add numerical values such as coefficients.

6. Discussion: P11, L356–357, L375–377: Please present this statement in the Results.

7. Conclusion: It would be better to be more specific.

8. “Limitations and outlook” should be described as part of the Discussion.

9. Citation of references: In “Instructions for Authors,” the following was noted: “References must be numbered in order of appearance in the text (including table captions and figure legends) and listed individually at the end of the manuscript.” Kindly rectify your text accordingly.

Author Response

Dear Reviewer 3,

The authors gratefully acknowledge the constructive comments on our paper. Below we reply to your specific comments in the referee report. For your convenience, we replicate your comments in the report first (in black) followed by our reply (in red). In the revised version of the paper, our changes are found in red.

We highly appreciate your contribution to this study with all your constructive comments!

Best regards,

Andor Toni Cigu

Elena Cigu

Round 2

Reviewer 3 Report

  • The manuscript has been greatly improved and is now in good condition